# Calibration of Impairment Severity to Enable Comparison across Somatosensory Domains

**DOI:** 10.3390/brainsci13040654

**Published:** 2023-04-13

**Authors:** Thomas A. Matyas, Yvonne Y. K. Mak-Yuen, Tristan P. Boelsen-Robinson, Leeanne M. Carey

**Affiliations:** 1Occupational Therapy, School of Allied Health, Human Services and Sport, La Trobe University, Melbourne, VIC 3086, Australia; y.mak-yuen@latrobe.edu.au (Y.Y.K.M.-Y.); l.carey@latrobe.edu.au (L.M.C.); 2Neurorehabilitation and Recovery, Florey Institute of Neuroscience and Mental Health, Heidelberg, Melbourne, VIC 3084, Australia; 3Department of Occupational Therapy, St Vincent’s Hospital Melbourne, Fitzroy, Melbourne, VIC 3065, Australia

**Keywords:** stroke, measurement, touch, proprioception, haptic object recognition, impairment, neurorehabilitation, scaling, normalization, cross-calibration

## Abstract

Comparison across somatosensory domains, important for clinical and scientific goals, requires prior calibration of impairment severity. Provided test score distributions are comparable across domains, valid comparisons of impairment can be made by reference to score locations in the corresponding distributions (percentile rank or standardized scores). However, this is often not the case. Test score distributions for tactile texture discrimination (*n* = 174), wrist joint proprioception (*n* = 112), and haptic object identification (*n* = 98) obtained from pooled samples of stroke survivors in rehabilitation settings were investigated. The distributions showed substantially different forms, undermining comparative calibration via percentile rank or standardized scores. An alternative approach is to establish comparable locations in the psychophysical score ranges spanning performance from just noticeably impaired to maximally impaired. Several simulation studies and a theoretical analysis were conducted to establish the score distributions expected from completely insensate responders for each domain. Estimates of extreme impairment values suggested by theory, simulation and observed samples were consistent. Using these estimates and previously discovered values for impairment thresholds in each test domain, comparable ranges of impairment from just noticeable to extreme impairment were found. These ranges enable the normalization of the three test scales for comparison in clinical and research settings.

## 1. Introduction

The aim of this Special Issue is to advance the neuroscience of touch and recovery of somatosensation after stroke. Somatosensation includes different modalities, such as touch and proprioception, as well as the processing and recognition of fused information across somatosensory modalities, such as with haptic object recognition [1,2]. As clinicians and scientists, we frequently encounter people who experience disruption to one or more somatosensory modalities, such as following a stroke [3,4,5]. A challenge in advancing our understanding for clinicians, researchers and survivors of stroke alike is being able to answer questions such as: what is the extent of disruption to one or more modalities, and is the nature and severity of impairment comparable across modalities? Addressing such questions has implications for the person’s understanding of somatosensory loss and the potential for recovery and rehabilitation [6,7,8,9,10,11]. 

This investigation engages the problem of calibrating scales of impairment in three domains of somatosensory function to permit comparison across domains in level and change in impairment. Quantification of impairment that is comparable is an issue fundamental to multiple scientific and clinical aims and relevant beyond the specific focus of this investigation, which is concerned with somatosensory impairment. Health professionals and researchers commonly wish to report on the severity of impairment of a particular function and compare it to related functions [6,7,10,11,12,13,14,15]. Whether an individual or a group is more impaired in one domain than another or whether an intervention that affects multiple domains produces a larger effect in one domain than another are questions that cannot be validly addressed without bringing domain-specific scales of measurement to a common base.

Scales of impairment severity require an origin. Typically, this is defined by a lower performance threshold in the unimpaired population, i.e., the criterion of abnormality [9,11,16]. Categorical judgments about the presence or absence of impairment can be made once a low score limit for performance by the healthy population is established. The severity of impairment can then be ordered within a given domain. It is common practice, in research and clinical settings, to use the units of test performance as an interval scale of impairment severity [9,17]. However, to obtain scales of impairment that permit a valid comparison of impairment quantity and severity, it is necessary to obtain scales with comparable units. 

Statistical methods that create unit-free scales, such as percentile rank scores or standardized scores, are sometimes employed to relate data from different scales of measurement. Standardization defines the fundamental scale unit as the standard deviation of the distribution of impaired scores. Instead of using the mean of the distribution, as in Z-score transformations, the test score for impairment threshold performance could be set as the origin of the impairment scale. In this scenario, the scale unit remains the standard deviation defined by the distribution of scores of the impaired population. Both percentile rank and standardized scores become unit-free values and promise the possibility of comparable scores across different test scales. 

However, these methods do not frame scales of measurement in terms of matching quantities of impairment directly, as, for example, do the Centigrade and Fahrenheit scales for temperature. Centigrade and Fahrenheit scales have matching quantities of heat, freezing and boiling points of water, and can be readily transformed into each other. 

The comparability of two unit-free scales obtained by standardization or percentile methods breaks down when the assumption of isomorphic (i.e., same shape) distributions is not valid. Figure 1 clarifies the problem using omniscient models. Hypothetical distributions of impairment for three domains are plotted. Since impairments are the manifestations of pathology as demonstrated by system function [18], zero is set to the threshold of impairment based on ability scores demonstrated by the healthy population in each domain. Such impairment thresholds are routinely employed in neurorehabilitation scales [9,17,19]. It would be possible now to transform raw scores to a unit-free scale by using the impairment threshold as an origin and dividing the difference between the test score and threshold score by the standard deviation for each domain. This method would not be problematic if distributions were isomorphic. However, Figure 1 shows why this breaks down when distributions are heteromorphic.

In the theoretical model depicted (Figure 1), zero represents the threshold of impairment (just noticeably impaired performance), while ten represents the maximum true impairment for each domain (a value which sometimes is not known). In this example, the three domains are assumed to have different distributions of impairment: exponential, uniform, and quadratic. An exponential function may arise when the mildest impairments are most frequent, with the probability of occurrence decreasing as severity increases according to an exponential function. A uniform function suggests all levels of impairment (mild, moderate, and severe) occur with equal likelihood. A quadratic function could arise if two types of system damage are hypothesized, each causing a characteristic extent of impairment but with some variation in degree, such as a low degree of damage with mild impairment in function for one type and a marked disruption of neural networks for the second type, resulting in severe impairment. In this scenario, the resulting distribution of impairment is bimodal, with modes at minimum (0) and maximum (10) impairment.

Figure 1a illustrates the probability density functions (PDFs) of the three distributions and shows that they have different standard deviations when scaled to reflect true impairment. This challenges the method of rescaling original test scores into standardized units using a single marker (e.g., the threshold of impairment or the distribution mean) and the standard deviation as the unit of measurement. The smallest standard deviation occurs for the exponential distribution (*σ* = 2). The uniform distribution has a standard deviation that is 144% of that for the exponential, a marked difference. The bimodal U-quadratic distribution has a standard deviation that is almost double (193.5%) that of the exponential and 134% of the uniform. Comparisons across such domains via standardization, when the standard deviation becomes the fundamental unit for each scale, would thus introduce substantial mismatches on the 0–10 scale of actual impairment. 

Figure 1b depicts the corresponding cumulative density functions (CDFs) of the three distributions, i.e., the cumulative probability of scores as a function of impairment magnitude. The three trajectories on this graph permit comparisons of the evolution of percentile scores for the three scenarios. While the 0, 50th, and 100th percentiles for the uniform and U-quadratic distributions are identical, all other values are discrepant, the discrepancy being in opposite directions and increasing in magnitude as they depart from the shared locations at the 0, 50, and 100 percentiles. The exponential model’s percentile values rapidly deviate from both other models. The discrepant form of the CDFs illustrates the problem of using percentile ranks as a method of recalibrating observed test scores to allow comparison across domains. Consideration of the form of each domain’s distribution of impairment is necessary.

This theoretical analysis shows the need to compare distributions of impaired scores in domains of interest. It also highlights the importance of locating comparable anchor values on test scales that operationalize impairment in different domains. Analogously to the creation of temperature scales, at least two comparable anchor values are an initial requirement for tests that operationalize impairment in each domain: the threshold of impairment (0 on Figure 1 abscissa) and comparable points of extreme impairment (10 on the abscissa of Figure 1).

The aim of the present study was to investigate a valid comparison of scores across three somatosensory domains of interest: touch discrimination, limb position sense, and haptic object recognition. We chose to focus on discriminative somatosensory function given the presence of ongoing impairment in discriminative functions [3], the negative impact on function [7,20,21], and the targeting of restorative approaches to retaining discriminative functions in rehabilitation [22]. Focus on the distal upper limb is also consistent with rehabilitation targets [6,23] and with feedback on the value of training somatosensation of the arm and hand, as reported by survivors of stroke [24]. Importantly, we have a history of evaluation and investigation involving the three modalities of texture discrimination, proprioceptive discrimination, and haptic object recognition [13,25,26,27,28,29,30,31,32,33]. Specifically, the Tactile Discrimination Test (TDT) [34,35], Wrist Position Sense Test (WPST) [36], and functional Tactile Object Recognition Test (fTORT) [37] have been used repeatedly with survivors of stroke in the past, providing a body of evidence from which to explore the issue of cross-modal calibration of existing measures. The measures are quantitative; standardized; have face validity, small scale intervals, age-matched normative standards, unidimensional scale, and strong psychometric properties. Pooling of data across data sets also provided access to relatively large cohorts of survivors of stroke that had been well phenotyped in relation to their somatosensory function, as well as background characteristics. Thus, these measures are both relevant to be used for this investigation and quality data were available. 

We investigated the shapes of distributions of test scores obtained from impaired populations of stroke survivors across these three domains of somatosensory function. Comparable loci of extreme impairment were also investigated. Both raw test score scales and scales normalized via two comparable loci of impairment severity, i.e., a threshold of impairment and a location for ‘extreme impairment’, were investigated. Impairment thresholds were available from previous investigations conducted by our group [34,36,37] and recently updated using pooled samples reported in the current study and related studies [35]. In the present context of somatosensory impairment, the function of persons who lack sensation following stroke (i.e., insensate) in the defined domain was a logical candidate for defining extreme impairment. To assess comparable loci for extreme impairment, the present investigation conducted simulations and theoretical analyses designed to model the likely distribution of test scores from a responder who is completely insensate in the relevant domain.

## 2. Materials and Methods

### 2.1. Study Design

This study involved pooled baseline assessment data of stroke survivors from 6 studies: Discriminative validity study [3,34,36,38]; SENSe (Study of the Effectiveness of Neurorehabilitation on Sensation) [23]; additional testing linked with the National Institute of Health (NIH) Toolbox study [13]; CoNNECT (Connecting New Networks for Everyday Contact through Touch) study [33,39]; IN_Touch (Imaging Neuroplasticity of Touch) study [31,40]; and SENSe CONNECT study [41]. Baseline data were extracted and pooled across studies. Test score distributions for tactile texture discrimination (*n* = 174), wrist proprioception (*n* = 112), and haptic object identification (*n* = 98) of stroke survivors were included. Inclusion and exclusion criteria were similar across studies.

### 2.2. Participants 

Participants recruited to the current study had a diagnosis of stroke and were assessed as having upper limb somatosensory impairment in at least one of three somatosensory modalities: tactile discrimination, proprioception, and/or haptic object recognition. They were medically stable, able to give informed consent and comprehend simple instructions. Participants were excluded if there was evidence of unilateral spatial neglect (based on the symbol cancellation test [42] and/or line bisection test [43]), previous history of other central nervous system dysfunction, and/or peripheral neuropathy. Participants from CoNNECT and IN_Touch studies were right-hand dominant, first episode infarct, no brainstem infarct, and eligible for MRI. All participants gave voluntary informed consent, and ethical approval was granted by the Human Ethics committees of participating hospitals and La Trobe University, Australia. 

### 2.3. Procedure and Assessments

All participants were assessed at baseline, i.e., from 2 to 613 weeks post-stroke. The following three somatosensory assessments were administered.

#### 2.3.1. Tactile Discrimination Test (TDT)

The TDT measures the ability to discriminate differences in finely graded texture surfaces using a three-alternative force choice design [34]. The participant is required to tactually explore the sets of triplet texture grids with their preferred finger (index or middle) and indicate the one that is different. The TDT25, the current version of the test used, has 25 trials, i.e., half the number of trials of that originally used by Carey et al. [34], to reduce duration and demand on the responder. On each trial, the TDT25 employs a standard grid, with a spatial interval of 1500 µm (micrometer) between the leading edge of one ridge to the next, and one of five comparison texture grids (1550, 1700, 2100, 2600, or 3000 µm). There are five trials per comparison level, and the number of correct responses out of 5 is recorded. The test has high test–retest reliability (r = 0.92), good discriminative test properties, and normative standards [3,34]. 

The TDT25 score used in this study is an area score based on an update to the original test scoring, as described below. The TDT25 area score uses an empirical approximation of the area under the psychometric function as the performance score (see Figure 2). The discrimination limen, originally defined as the percent spatial increase (PSI) score value [34,38], is no longer used because our group has found some stroke survivors respond with a psychometric function that is not fully characterized by the location of the inflexion point of a sigmoid function, i.e., the limen. An undistorted sigmoidal function is observed in both unimpaired responders and in a substantial proportion of stroke survivors [34]. The area score was introduced because the sigmoid function in some stroke survivors is distorted: it may show correct response rates above chance expectation but not reach the level that defines a limen even at the largest differences in texture, or it may plateau at a correct response probability below 1.0, or even below that required to define the limen. These observations suggest that sole reliance on the limen does not fully capture the impairment in performance due to stroke. 

To overcome this failure of the limen as an index of impaired performance, without undermining applicability in clinical settings, test results can be quantified with a score based on the area under the polygon determined by the distance between stimulus comparison values, and the correct responding rates out of the 5 trials for each comparison value [31]. This ‘psychometric polygon’ (Figure 2) is a nonparametric empirical approximation of the area under a continuous, curvilinear psychometric function. Unlike fitting a continuous curve for the psychometric function, this empirical approximation does not assume a (sigmoid form) model nor require complex model choices or model fitting computation (an impractical goal for clinical settings, responders with impaired somatosensation, and limited data sets imposed by these testing conditions). This approximation copes with atypical response patterns where a limen is not achieved or where performance deterioration, additional to what is reflected in the limen location, is evident in abnormal plateaus or in other supra-limen performance. The area score captures performance influences throughout the stimulus range. It is reminiscent of a weighted average of the correct response rates for each of the comparison stimulus differences, where the weights depend on the distances between adjacent comparison values (Figure 2) and whether a stimulus is at one of the two extremes of the stimulus set. Hence, the TDT score will be referred to as the TDT percent maximum area (PMA) score.

#### 2.3.2. Wrist Position Sense Test (WPST) 

The WPST measures the ability to perceive wrist position whilst placed in a splint within a boxlike apparatus with vision occluded [36]. In each of the 20 trials (WPST20), the test involves matching a predetermined imposed wrist position from the flexion-extension range using a pointer aligned with the axis of wrist movement and a visible protractor scale. The assessor applies a preselected position to the wrist while the responder is instructed to remain still in the hand splint. The responder then uses the other hand to indicate, via an aligned pointer, the proprioceptively sensed position on the response protractor scale. This test has normative standards for adults (~40–85 years), high retest reliability (r = 0.88 and 0.92) and good discriminative validity when used with adult stroke survivors [3,36]. A detailed administration description is provided by Carey et al. [36]. The overall average error score is the mean absolute deviation (MAD) between imposed stimulus angles and corresponding response angles.

#### 2.3.3. Functional Tactile Object Recognition Test (fTORT) 

Impairment in haptic object recognition was operationalized using the fTORT [37]. On each trial, with vision occluded, participants feel one object selected from 14 object sets. The responder then uses a poster picturing 42 objects, comprised of 14 sets of three objects, to indicate the object they feel. Each object set comprises a target object, another object that differs in one sensory attribute only, e.g., weight, and a third object that differs in two sensory attributes, e.g., weight and shape. Only common, everyday objects that can be readily manipulated are included. Selecting the correct object scores 3, while an error in one sensory attribute, e.g., weight or volume of the object, scores 2. Selecting the object within the functional triad that differs in two sensory attributes scores 1. Selecting any of the 39 objects outside the functional triad scores 0. The test forms a well-behaved unidimensional scale, demonstrates excellent internal consistency [37], and has good discriminative test properties [44]. The overall test score for the fTORT ranges from 0 to 42, i.e., item scores from this unidimensional are added into a single total score [37], with 42 indicating the best performance.

### 2.4. Data Analysis

To characterize the statistical properties for the distributions of responses expected from responders forced by insensate status to guess on each of the three tests (referred to here as ‘extreme impairment’), theoretical and several simulation analyses were conducted. All simulations were coded using RStudio 1.3.1056 [45]. Some simulations were replicated with additional assumptions that incorporated potential response biases. The analyses to establish expected distributions of responding from completely insensate responders are described in the subsections that follow. 

IBM-SPSS V.28 (Statistical Packages for Social Sciences, Chicago, IL, USA) was then employed to construct frequency histograms of the observed TDT25, WPST20, and fTORT scores and to evaluate the statistical properties of these empirical distributions. Normalized scales for each domain were obtained using the loci for extreme impairment identified by the theoretical and simulation analyses described above and the impairment threshold loci identified in previous reports [34,36,37] and updated based on larger pooled samples available to the current and related studies [35]. Normalization set the location of extreme impairment at −100 and the threshold of impairment at 0 for each domain. The TDT25 and fTORT scores decrease as impairment increases, but the reverse is true for the error scores of the WPST20. Therefore, the latter was first multiplied by −1 to reverse the direction of the scale. Cumulative distributions for each of the three normalized data sets were obtained. TableCurve 2D V5.01 software (Systat Software Inc., Chicago, IL, USA) [46] was employed to fit cumulative distribution functions (CDFs) for the observed cumulative distributions. Kolmogorov–Smirnov tests were applied to assess nonparametrically the statistical significance of differences between cumulative distributions.

#### 2.4.1. Simulation of Expected Distribution from an Insensate Responder Using the TDT25

Analyses aimed to discover the distribution of area scores likely to occur when guessing at random and with some positional response biases. Theoretically, the long-run expectation from pure guessing of the position for the odd stimulus in a triplet set is 1/3, irrespective of the texture difference between the odd stimulus and the twin pair. With a probability of 1/3 correct on all five comparisons comprising the TDT25, the area scores would be 1/3 of the maximum. Therefore, the theoretically expected score for the centre of the distribution of insensate responders is 33.3%. 

Obtaining a full description of the distribution is more difficult. According to the combination with replacement theory, there are 252 possible response combinations for which to calculate area scores, making the area computations very demanding. We therefore employed a simulation approach, which presents a simpler coding problem. A computer was coded to select a response position from 1, 2, or 3 at random. The correct response position was stipulated by imitating the actual test protocol. This procedure also had the advantage of permitting simulations that included some response biases in addition to the one based on completely random response selection. A score of 1 was logged if the response position coincided with the stipulated correct position. This was repeated for 25 trials (5 comparisons × 5 trials each) as in the actual test. The number of correct responses (0–5) for each of the five comparison values was logged, and the area for that response pattern was calculated and recorded. This sequence of steps formed a loop that was repeated 10,000 times. Seven simulation scenarios were evaluated: responding completely at random (equiprobable likelihood for the three stimulus positions); responding with a central bias or a strong central bias (50%, or 66.67% of responses for the middle stimulus); responding with a left or with a right bias (50% of responses for the specified lateral stimulus); and with a strong left or right bias (66.67% of responses for the specified lateral stimulus).

#### 2.4.2. Simulation of Expected Distribution from an Insensate Responder Using the WPST20

Theoretical analysis of what can be expected from a completely insensate responder on the WPST20 must also begin with the assumption that the responder is forced to guess a position in the range of movement of the wrist. Although the apparatus protractor scale portrays a range of ±90° around the neutral starting position of 90°, the literature suggests a healthy movement range of 60°–75° for both flexion and extension [47,48]. Several variables such as aging and degenerative conditions are also likely to further restrict that range for various subpopulations. Therefore, a realistic starting assumption for the combined range of movement is 140°–150°. We further assumed that people have little attentive experience with passive wrist flexion-extension range of movement. We also assumed that memory models driving proprioception, particularly after the loss of sensory input, would be dominated by prior active movement experience, and thus, the mental model for insensate responding may have a range less than 140°–150°. In addition, the literature on response biases over a range of estimation tasks has identified an end aversion bias (e.g., [49]). Finally, the actual test angles used were defined within a comfortable range of flexion-extension movement, considering potential restrictions in range due to age and increased tone after stroke, i.e., 100° total range, 65° for flexion, and 35° for extension [36]. Taking these assumptions and actual test angles into account, it seemed reasonable to hypothesize that the distribution of responses in the WPST20 given over many trials by an insensate person is likely to have a range of less than 70° of flexion or 70° of extension. 

Three simulation studies were conducted. Each selected 10,000 random integers between stipulated maximum and minimum angular values. For the first simulation model, the selected limit for responses in the extension range was 45°, while for the flexion range, it was 70°. For the left wrist WPST20, this implied creating random responses between 45° (90° − 45° = 45°) and 160° (90° + 70° = 160°) for each of the 20 wrist angle positions set via imposed movement, as stipulated by the WPST20 protocol. Obtained response integers deviated from each corresponding target value, and error scores were calculated for each of the 20 target angles to replicate the WPST20 procedure. The statistical distribution of the resulting 10,000 MAD error values was then obtained. This simulation protocol was repeated for a more restricted response range of 55°–155° (consistent with the actual testing range) and for one equivalent example using test angles from the contralateral side (right wrist), which ranged from 25° to 125°. The distribution of random integers created by the code for each of the 20 target angles should be uniform if the code performed correctly, as random ‘responses’ were programmed to be equiprobable within the stipulated range. This was verified.

#### 2.4.3. Evaluation of Expected Distribution from an Insensate Responder Using the fTORT

For the fTORT, given a completely insensate responder who chooses at random among the 42 pictures, the calculation of the probability of correct and partially correct scores according to the calculus of probability and the cumulative binomial distribution was carried out.

## 3. Results

### 3.1. Characteristics of the Sample

Demographic and clinical characteristics of the pooled sample for each of the somatosensory domains are presented in Table 1.

### 3.2. Distribution of Scores Expected from an Extremely Impaired Responder on the TDT25

The obtained distribution of area scores for extreme impairment is presented in Figure 3. As expected, theoretically, the histogram approaches Gaussian form with a mean of 33.3% (33.31) and a median of 33.29, with values very close to the theoretical expectation. The simulation also revealed parameter values that would have presented much greater difficulty to estimate by theoretical means. For example, the standard deviation of this approximately symmetrical distribution (skew index of 0.174) is 10.54. Given the approximately Gaussian form of the distribution, this standard deviation implies that area scores of substantially larger or smaller values than 33.3% will be observed from respondents functioning purely by guessing. The table below, which summarizes observed percentiles, also shows that remarkably small areas, substantially below 33.3% of the maximum, can occur purely through guessing in the TDT25. For example, the smallest 10% of area scores obtained purely by guessing will be 20 or less, which is substantially below 33.3. Similarly, areas well above 33.3 that are in the forty-point range between 33.33 and the impairment threshold of 73.1 PMA can occur purely by guessing. 

The results of Figure 3 are based on a simulation of a respondent guessing at random between the three stimulus locations without any positional bias. Additional simulations were conducted to investigate the effect of a bias to respond more often in the central position of a stimulus triad, to the left, or to the right. Two intensities of bias were investigated: (a) a ‘milder’ bias, which shifts the response probability from 33.33% of the time to 50% (‘milder’ bias); (b) a stronger bias, which shifts the response probability to 66.67% of the time, i.e., twice as often as would occur through unbiased guessing. The results are presented in Table 2.

The responses confirm that response biases can shift the distribution of scores, although the consequences seem mild, with the possible exception of a strong central bias, which tends to cause an increase in the average area score from 33.3 to 39. Strong biases to either side produced a mild decrease in the distribution mean and variability. All distributions displayed relatively large dispersions.

### 3.3. Distribution of Scores Expected from an Extremely Impaired Responder on the WPST20

All simulation runs displayed distributions with bell-shaped histograms that followed the predicted values from continuous normal distribution well, as expected from the central limit theorem (Figure 4). Error score distributions had means located in the range of 33° to 36° average error, which implies that the statistically expected value for extreme error scores in the WPST20 is in that range. Simulation 1, which had the largest range of possible responses, showed the largest distribution mean and standard deviation in accord with the theoretical expectation. Standard deviations ranged from 4.94 to 5.42, yielding coefficients of variation ranging from 0.147 to 0.15. As theoretically expected, the narrower ranges of movement yielded smaller distribution means and smaller distribution standard deviations. 

### 3.4. Distribution of Scores Expected from an Extremely Impaired Responder on the fTORT

Figure 5 shows the calculated proportions of scores expected to occur under random response selection for scores 0–6. The mode of the distribution is a score of 0, with a probability of 0.354. The likelihood of higher scores declines rapidly in this strongly skewed distribution. More than 95% of all possible outcomes are covered by scores of 6 or less; hence, specific probabilities for higher scores were not estimated. Total fTORT scores of 0 and 1 together comprise 48.15% of the total distribution, close to the median. As scores of 2 comprise 14.84% of the distribution, the results suggest an anchor for extreme impairment between the real discrete scale score possibilities of 1 and 2, comparable to the medians of the distributions for the other two tests, which have highly symmetrical distributions of scores obtained from random responding, where means and medians do not markedly diverge as for the fTORT score distribution. 

### 3.5. Morphological Comparison of Distributions for Impaired Tactile Discrimination, Wrist Position Sense and Haptic Object Recognition

Figure 6 shows histograms for each of the three observed samples of scores when scaled in original scale units (left column of panels) and after transformation into normalized scale units (right column of panels). Inspection of the histograms suggests different distributions apply to the three domains. The histogram for TDT25 scores suggests a truncated peak distribution with its center of gravity somewhere in the middle of the impaired performance score range. In contrast, the distribution of WPST20 scores resembles an exponential function or perhaps a lower tail segment of a strongly asymmetric peak distribution (e.g., Gaussian) whose peak lies unobserved in the unimpaired range. The asymmetry would need to be strong given the much shorter range (less than 11°) for mean error scores in the unimpaired performance range, which appears to be less than a third of the impaired score range. The fTORT histogram seems consistent with a bimodal U-shaped distribution, with modes at the extreme impairment end and at the just-impaired end. A plausible alternative hypothesis is for a uniform distribution, with frequency variations that suggest the two modes and trough in the mild-to-moderate range of scores are sampling errors. The hypotheses suggested by inspection of these histograms imply different distribution forms do apply to the three domains of measurement.

Normalization of the three raw scales should not and did not alter the overall pattern that suggested different distribution forms apply to the three domains of measurement. Normalization did, however, suggest that the TDT25 scale has a larger proportion of scores at values more extreme than −100 compared to the other two scales. The fTORT distribution truncates at 0 on the raw score scale, and little data exist at raw scores of 0 and 1, which are the only two scores possible below the value of 1.5 used to fix the normalized point for extreme impairment of −100. Similarly, the WPST20 scores below −100 are few (3.6%) and sparsely distributed, in accord with the vanishing probability expected from the tail of an exponential distribution. 

To confirm the hypotheses suggested by the histograms, plausible CDFs were fitted to each of the three data sets (Figure 7). Kolmogorov–Smirnov Z (two-tailed) tests confirmed that the cumulative distributions of the normalized scales were significantly heteromorphic for each of the three pairwise contrasts: Z_K-S_ = 4.149, *p* < 0.001 when comparing the TDT and WPST distributions; Z_K-S_ = 1.906, *p* = 0.001 for the fTORT versus TDT comparison; and Z_K-S_ = 3.083, *p* < 0.001, for the WPST versus fTORT comparison. Figure 7 shows that fitted CDFs for the three measurement domains conformed to the distinct theoretical forms suggested by inspection of the raw data histograms. A cubic polynomial function best fitted the cumulative distribution for the fTORT data (Figure 7). In contrast, the TDT25 cumulative distribution conformed better to a cumulative Gaussian function, which has the inverse pattern of curvature to that of the cubic polynomial fitted to the fTORT cumulative distribution (Figure 7). In contrast to both of these, the WPST20 data were fitted well by an exponential function, which curves in only one direction (Figure 7).

### 3.6. Morphological Comparison of Distributions for Impaired Tactile Discrimination, Wrist Position Sense, and Haptic Object Recognition Adjusted for Time Post-Stroke 

We conducted post hoc analyses to adjust for the stroke onset time, i.e., stroke latency. Using the fitted models, we obtained adjusted scores that discount the effect of stroke latency and then constructed histograms to discover if the adjusted scores still show different distribution shapes. Test scores collected at baseline were adjusted for the effect of latency of the test post-stroke for the TDT25 and fTORT. Weak but statistically significant logarithmic relationships between test latency and the TDT25 score (r^2^ = 0.104) and between test latency and fTORT scores (r^2^ = 0.067) were removed by fitting logarithmic functions between test latency and each of the test score samples, saving the unstandardized residuals and adding the model predicted mean value to these residuals, thereby reconstituting original scores minus the effect of test latency. The original WPST20 score distribution was not adjusted as the effect of latency was negligible (r^2^ = 0.009) and not statistically significant. The three histograms retained the differences in shape indicated by the untransformed score distributions, showing that even after discounting the impact of test latency, the three test score distributions remain heteromorphic.

## 4. Discussion

Our aim was to bring performance from different somatosensory domains to equivalent scales of the extent of impairment to facilitate a valid comparison. Distributions of scores in the three domains investigated were found to be heteromorphic. This implies that comparison of the severity of impairment across the three domains using percentile score scales or standardized score scales (a common practice) is not appropriate, as severity scores would refer to different loci in the range between just noticeably impaired performance and extremely impaired performance (Figure 1). A solution that avoids this problem is to normalize raw score scales using empirically identified loci for the threshold of impairment and extreme impairment set at comparable values. The proposition that the severity of sensory impairment grows between an origin at the borderline with low performance in the healthy population until it reaches an extreme value defined by the response capacity of an insensate responder: (a) seems logical as a range of impairment severity; (b) applies to all sensory dimensions; (c) is consistent with the accepted definition of impairment severity [18]; (d) does not introduce scale units that explicitly assume isomorphic distributions of the dimensions to be compared.

The major value of the normalized scales defined is that they establish equivalent ranges of impairment across different domains. Loci for the threshold of just noticeable impaired performance (criterion of abnormality) were based on the performance of age-matched neurologically healthy individuals, updated using pooled samples. Loci of extreme impairment identified from our theoretical and simulation analyses were consistent with values suggested by distributions of test scores from stroke survivors in each of the three somatosensory domains examined, providing independent empirical confirmation. These convergent results enable initial estimates for a marker of extreme impairment in each of the three domains investigated. The simulation studies support the location of comparable extreme impairment markers at a score of 33.3 PMA for the TDT25 score, 36° average error for the WPST20, and 1.5 for the fTORT raw scale. With two ends of the scale—just noticeable impairment and extreme impairment—now comparably identified across the different domains, we have established equivalently normalized scales for future comparisons. This approach of identifying quantities of impairment directly aligns with the example of Centigrade and Fahrenheit scales for temperature and with the recognition that when interpreting test scores, “scores cannot be ‘impaired’, only a function can be impaired” [19].

The anchor values that enabled comparable normalization should also assist clinical interpretation, provided an allowance for measurement error is made [34,36,37]. For example, working with the largest available sample of impaired stroke survivors, we found the norm-referenced just noticeable impairment threshold for the TDT25 was 73.1 PMA [35]. The 33.3 PMA extreme impairment marker suggested by the results for the TDT25 provides an additional perspective. For the WPST20 test, the just noticeable impairment threshold was considered as 11.3° average absolute error, with 36° being the best estimate of extreme impairment. For fTORT, the just noticeable impairment threshold was 39.5, and 1.5 was the anchor of extreme impairment. Thus, scores less than 39.5 and greater than 1.5 quantify impairment within an empirically defined severity range. 

The observed distributions for WPST20 and fTORT appear to be consistent with the loci for extreme impairment indicated by theoretical and simulation analyses. The exponentially decaying distribution for WPST20 scores is well into its tail at −36°, an extreme impairment locus close to those suggested by the simulations (Figure 4). The most extreme error scores on the WPST20 beyond −36° were all between −43° and −36°. This range is no further than 1.3 standard deviations away from a normal distribution centered on −36° according to the simulation studies. A normal distribution seems the most plausible hypothesis for sampling errors when measuring average error, and the simulations show that error scores between 36° and 43° are commonly expected (40% of the time) in conditions of extreme impairment. Sampling errors can thus readily account for the few observed scores more extreme than −36°. Similarly, the fTORT sample showed 3% of data to be between 0 and 1, with the low-value mode of the bimodal distribution in the class interval comprising scores of 2 and 3. The low mode is close to the extreme impairment locus between scores of 1 and 2 suggested by the theoretical model (Figure 5). The proportion of observed scores at 0 (2%), 1 (1%), 2 (3. 1%), and 3 (6.1%) is readily accommodated by the theoretical analysis, which found 48% of insensate responders are expected to obtain total scores of 0 or 1. Thus, an extreme impairment locus above 1 and not higher than 3 is supported by the observed data. A range of 1.5 to 3 represents a tolerable uncertainty interval of 4% relative to the distance from the impairment threshold. 

The raw score distribution for the TDT25 showed a thicker tail below the extreme impairment mark (33.3) indicated by simulation studies, in contrast to the distributions for the other two domains. The presence of a relatively high frequency of extreme impairment in tactile discrimination is consistent with clinical observation and with scores on the original TDT with 50 trials [34]. Detailed evaluation of the simulation results suggests the observed distribution can be explained by sampling error in a test with only 25 trials. According to the simulation results for unbiased random response selection (Table 2), the subpopulation of such responders is expected to have a normal distribution with the fifth percentile at 16.55 PMA. The observed distribution of TDT25 scores from stroke survivors (Figure 6a) found 5.7% of scores at or below 16.55 PMA. The observed score distribution may also be regarded as a sample from a compound population composed of subpopulations (subtypes) of stroke survivors that range in true score values from 33.3 to higher values. This implies that if the histogram of observed scores for the TDT25 (Figure 6) is regarded as a compound distribution, the 27.7% of observed scores below 33.3 can be accounted for by the sampling error of pooling people with true scores of 33.3 and people with higher true scores. This conclusion implies there is no inconsistency between the observed distribution of TDT25 scores and the theoretically expected anchor for extreme impairment at the unbiased guessing expectation of 33.3. This argument is also another indication of the value of undertaking the reported simulation studies: using a conjunction of simulation analyses and observed score distributions to estimate an extreme impairment locus is superior to reliance on observed score distributions alone.

Rescaling the different domains to comparable unit-free scales via standardization appears to introduce substantially greater error in calibrating task performance than doing so via normalization using two directly comparable performance standards available across the three domains: the impairment threshold and the locus of extreme impairment. Some error must be allowed for locating the corresponding impairment thresholds, although these magnitudes are likely to result in a small impact on the normalized range of impairment. For example, for the WPST20, the original criterion of abnormality was 10.5° average error [36], while the recent pooled data analysis found a criterion of 11.3° average error. In addition, the simulation studies for the TDT25 and WPST20 provide evidence that bias-driven variations did not create marked departures from the random response models. In comparison, the errors of scaling likely to arise from using the standard deviations of distributions are known to have different shape distributions that appear to exceed those that are likely from an erroneous location of the threshold and extreme impairment anchors for normalized scales. Using the normalization scale, the difference between the impairment threshold and extreme impairment in raw scale units is 2.4 times the standard deviation of the distribution of raw TDT25 scores, 3.6 times the standard deviation for the raw WPST20, and 3.06 times for the fTORT. In comparison, using standardized scores to compare across scales places extreme impairment on the WPST20 150% further away from the impairment threshold compared to the TDT25 and 118% further away compared to the fTORT. Extreme impairment for the fTORT is located 128% further away from the impairment threshold than for the TDT25. Thus, the errors from ignoring the effect of shape differences are larger than the errors likely from the alternative method (normalization), which relies on locating two anchors. 

The different shapes of the three distributions may contain insights about the impairment of function in the three domains after acknowledging that their forms can also be impacted by sampling processes and test construction factors. As the data for the three tests were mostly obtained from the same participants with similar inclusion criteria across studies, sampling factors seem an unlikely explanation for the observed heteromorphism of the three score distributions. Time post-stroke could influence scores within and across domains, contributing to differences in the shapes of the three distributions. Analysis of scores adjusted to remove the effects of time post-stroke found that the differences in the three distributions cannot be accounted for by any effect due to different latencies of testing in the three domains. 

Test properties could account for some of the observed differences in distribution shapes. The TDT25 has the lowest center of gravity of the three distributions (normalized scores) in the moderate-severe range (mean = −73.2; median = −76.8) and a substantial lower tail, suggesting the test could benefit from better discrimination in the severe to very severe range by including some easier test items, which would tend to move the distribution peak towards lower impairment scores and likely reduce the number of scores below the extreme impairment marker. Increasing the WPST20 test difficulty, e.g., through some masking or distraction procedure, might shift the distribution towards the more severe impairment range; however, this would seem to diminish rather than enhance the construct validity of the WPST20. The fTORT was designed to quantify impairment in the ability to haptically discriminate objects encountered in everyday function. The observed bimodal distribution seems a mild deviation from a uniform distribution, and item difficulties were not found to vary markedly in the test construction phase [37]. It is difficult to argue, therefore, that the distribution shape is determined by including inappropriate items or a bias towards items that are too difficult or too easy.

It is possible that neural processes and brain regions/networks that support somatosensory processing contribute to different impairment distributions. For example, the suggestion of subtypes of touch discrimination impairment and high frequency of extreme impairment appears to be consistent with clinical observations of marked impairment in touch detection (i.e., loss of protective sensibility based on monofilament testing) for many of the individuals with extreme touch discrimination impairment [23]. Some level of touch detection is required for texture discrimination, and this impairment may persist in tandem with extreme touch discrimination impairment. In addition, our meta-analysis of brain regions highlights the bilateral and lateralized representation of touch [50], which is consistent with the suggestion of subtypes of impairment with varying lesion locations. Further, chronic touch discrimination impairment has also been associated with disruption to fiber tracts remote from the lesion [33], which is consistent with the observation of persistent impairment and relatively high frequency of extreme impairment. 

In the comparison, the frequency distributions of discrimination of limb position, tested using the WPST, showed a low proportion with extreme position discrimination impairment, which is consistent with recent studies from other groups [5]. Proprioception is associated with brain activation in high-order sensorimotor cortices, including the right supramarginal gyrus, in healthy and stroke-affected participants [51,52], and persistent impairment is associated with damage to such regions [53]. It is possible that the involvement of regions beyond the primary somatosensory cortex and the involvement of both hemispheres provides some redundancy. Such redundancy may be capable of limiting the effect of neural damage on proprioception, resulting in a comparatively low frequency of severe errors. Further analysis of the association between clinical testing with the WPST and functional and structural brain changes is required. In relation to haptic object recognition, which is likely contributed to by both tactile discrimination and limb position sense, it is possible that the presence of higher frequencies for both mild and extreme impairment could reflect varying disruption to one or both somatosensory modalities and their supporting brain networks. Further investigation of these relationships is also needed.

The interpretation of the severity of impairment depends on the frame of reference. The development of normalized scales anchored to the impairment observed in completely insensate performers enables a perspective on the severity of impairment that is anchored to a natural extreme in the dysfunction of the somatosensory system and to the defined somatosensory task. If impairment is defined as the ability to perform a defined task, then equal distances on the task dimension could be considered equal distances on the impairment dimension. For example, if proprioception is the “ability to report the location of a limb in space”, then it appears logical that the difference between an error of 12° and an error of 8° is comparable to the difference between errors of 16° and 12°. Similarly, texture discrimination can be defined as the “ability to detect the difference between two textures”, and the difference in texture grids is objectively defined in distance units. Thus, calibration by normalization may be preferred as it directly identifies comparable locations on the respective dimensions of task performance that define the domains to be compared. It does not rely on comparing locations defined by the score distributions obtained from the pathological population. 

There are limitations to the current investigation. First, we chose to investigate the issue of calibration across domains using three somatosensory domains and three existing relevant measures that have been used repeatedly across studies with survivors of stroke, high-scale resolution, normative standards, unidimensional scale, and good psychometric properties. While our study demonstrated proof of concept evidence of heteromorphic distributions requiring alternative methods of comparable calibration (via normalization) across these three measures, we acknowledge that this is only one demonstration of the issue, and we encourage others to investigate this approach across other measures and domains of function. Second, the method of normalization proposed and tested lacks knowledge of the rate of progression of units across scales. To compare values between the just noticeable threshold and insensate extreme, it is necessary to assume the rate of progression is similar within the different domains. This may be the case if impairment directly relates to the response scale defined (as discussed above). An empirical evaluation of this was beyond the scope of the present study, which aimed to investigate three commonly used methods for facilitating comparison among pre-existing test scales. Nevertheless, normalized scales that have the two ends defined also have matched ranges.

Finally, our findings may also have relevance to other somatosensory measures and to other domains of impairment, such as motor impairment following stroke. The investigative approach illustrated by the present study can be applied to other scales of impairment when calibration is intended to facilitate cross-domain comparisons, provided the measures for the domains to be compared have been demonstrated to be unidimensional scales. A summation of test items has been proposed for some impairment scales in both somatosensory and motor domains (e.g., [54,55,56,57]). For example, some impairment scales sum test items obtained from multiple body parts, invoking an unobserved underlying quantitative dimension, a latent variable. Whether scores from items testing different body parts fit along a unidimensional scale and the proper relative weighting of score values from different body parts are issues that require resolution to claim a valid unidimensional scale. Once this issue of unidimensional scale is addressed, the issue of calibration across domains and which is the best method of addressing this, e.g., standardization or normalization, can be investigated. In taking the normalization approach illustrated by us, two anchor points are required. It is worth noting, however, that having a known location for theoretical extreme impairment on a domain scale is likely a useful point of reference for interpretation, irrespective of whether a normalization or standardization approach is most suitable. Finally, as indicated earlier, the issue addressed in our paper goes beyond somatosensation alone and may have relevance for comparison across motor domains or neurological domains more generally. 

## 5. Conclusions

Our findings indicate that for somatosensory domains, where the distributions of impaired scores differ but two comparable loci of impairment severity can be fixed for each domain, normalization should be preferred for cross-domain comparisons. Such normalization divorces scale calibration severity from the shape of the distribution of impaired scores and is compatible with the concept of a dimension of impairment [18]. This was the case for the three domains tested in this study, where distributions were strongly heteromorphic. Comparable loci for the threshold of impairment and extreme impairment appear to be achievable for all somatosensory domains.

## Figures and Tables

**Figure 1 brainsci-13-00654-f001:**
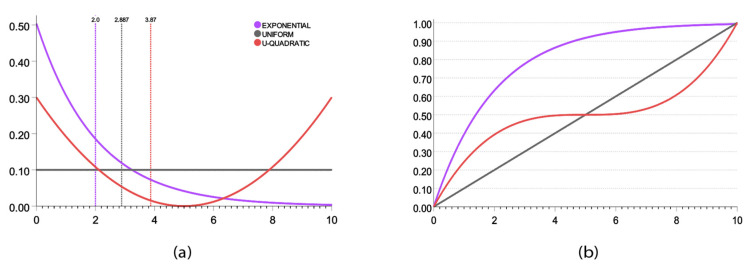
Probability density functions (PDF) and one standard deviation distances from origin for three hypothetical models of impairment distribution (**a**) and their corresponding cumulative distribution functions (CDF) (**b**). The three distributions follow Exponential (purple), Uniform (grey), and U-Quadratic (red) functions. The abscissa represents amount of impairment. The ordinate quantifies probability (**a**) or cumulative probability (**b**).

**Figure 2 brainsci-13-00654-f002:**
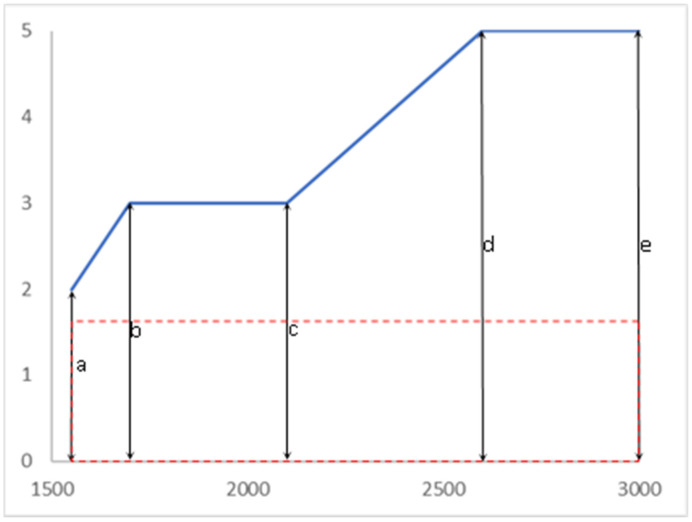
Illustrative example of computing an area score (percent maximum area, PMA) for the TDT25 from the polygon formed from the number of correct responses to each of the five comparison stimuli sets presented in the TDT25 protocol. The abscissa quantifies the stimulus dimension, while the ordinate quantifies the number of correct responses (out of 5) at each stimulus comparison level. The rectangle bounded by the red broken line illustrates the area that would be formed if each stimulus had a correct response rate of 1.66, i.e., the expected long-term rate of correct responding when guessing (NB In the three-alternative forced-choice protocol of the TDT25, the limen probability is (1.0 − 0.33)/2 = 0.67). The standard grid for each TDT25 stimulus anchor triplet is 1500 µm, and the five comparison stimuli are 1550, 1700, 2100, 2600, and 3000 µm (indicated as a,b,c,d,e on the abscissa). The overall polygon is the sum of four component polygons, each bounded by the respective correct response frequency and the corresponding subtending stimulus distance. In the illustrated example, the area of the overall polygon (TDT25 PMA score) is the sum of the component polygon areas, which is 78.97% of the maximum possible area that would be obtained if all comparisons had 5 correct responses out of the corresponding 5 trials.

**Figure 3 brainsci-13-00654-f003:**
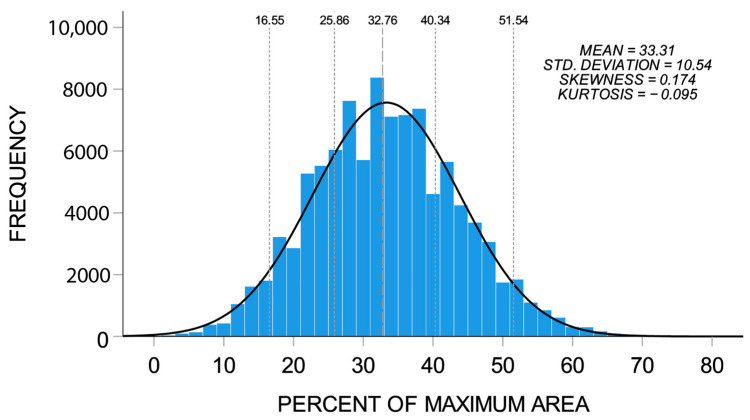
Distribution of TDT 25 PMA (Percent of Maximum Area) scores obtained from simulation study. Values for the 5th, 25th, 50th, 75th, and 95th percentiles are marked by the broken grey vertical grid lines.

**Figure 4 brainsci-13-00654-f004:**
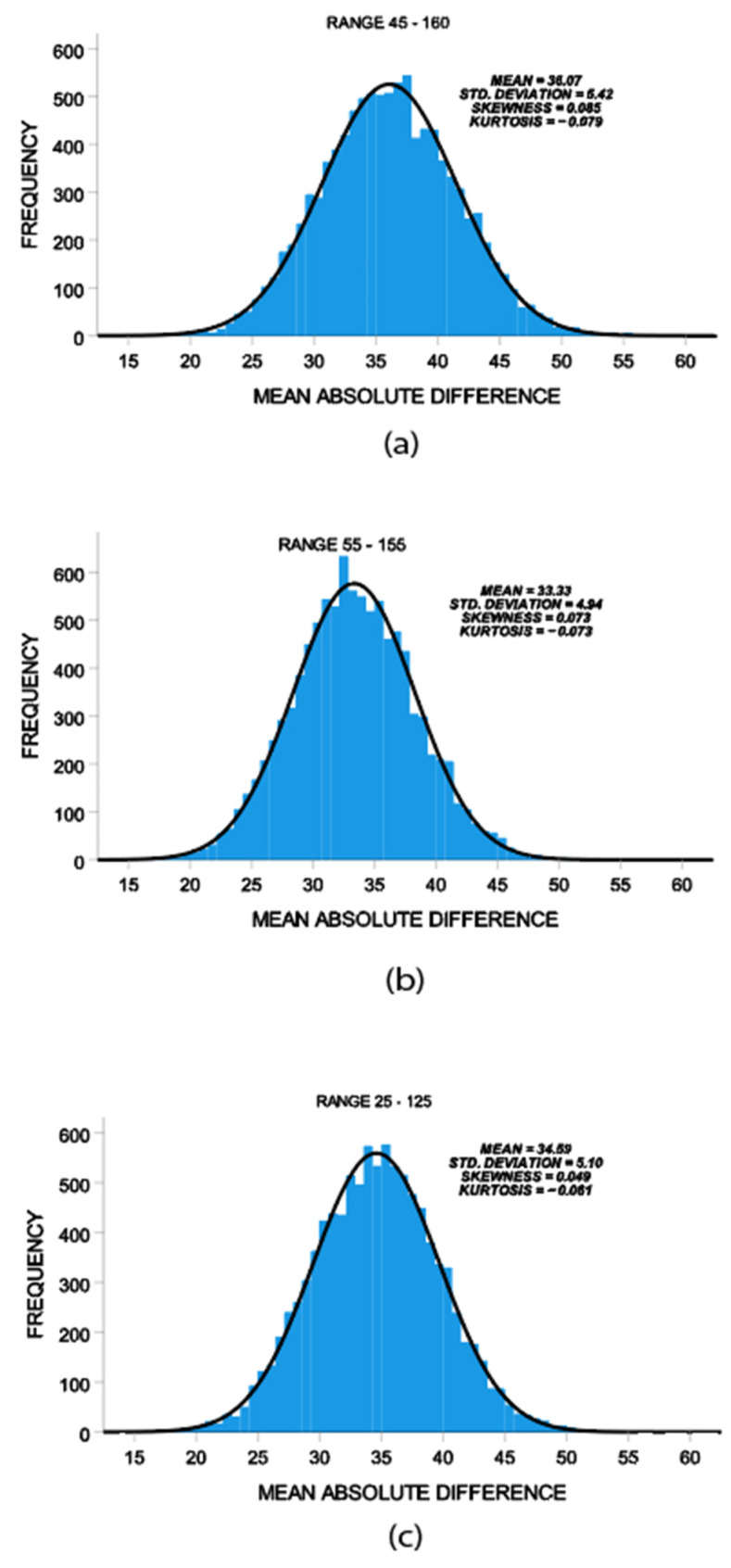
Principal results from three simulations of responses by a responder who guesses at random with equal probability from a given range of movement. Panels (**a**–**c**) show distributions for different response ranges. Panels (**b**,**c**) illustrate 100° range simulations for left and right wrists, respectively, demonstrating very similar distribution statistics.

**Figure 5 brainsci-13-00654-f005:**
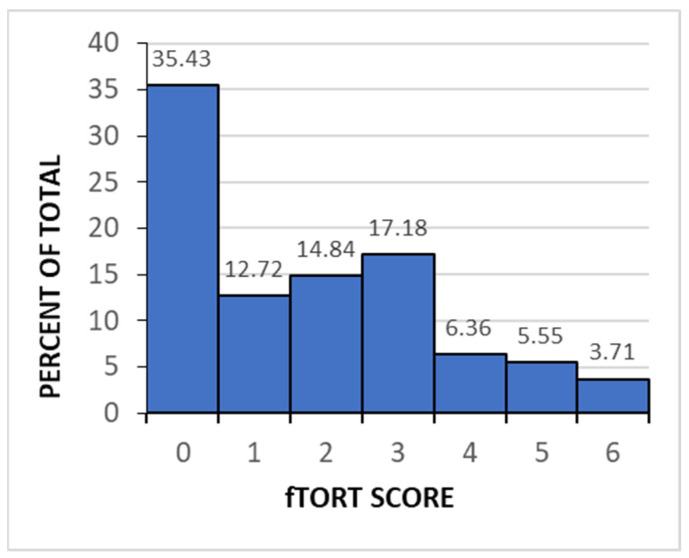
Distribution is expected for total scores of 0–6 in the fTORT if an insensate responder selects an object at random from the 42 items presented visually.

**Figure 6 brainsci-13-00654-f006:**
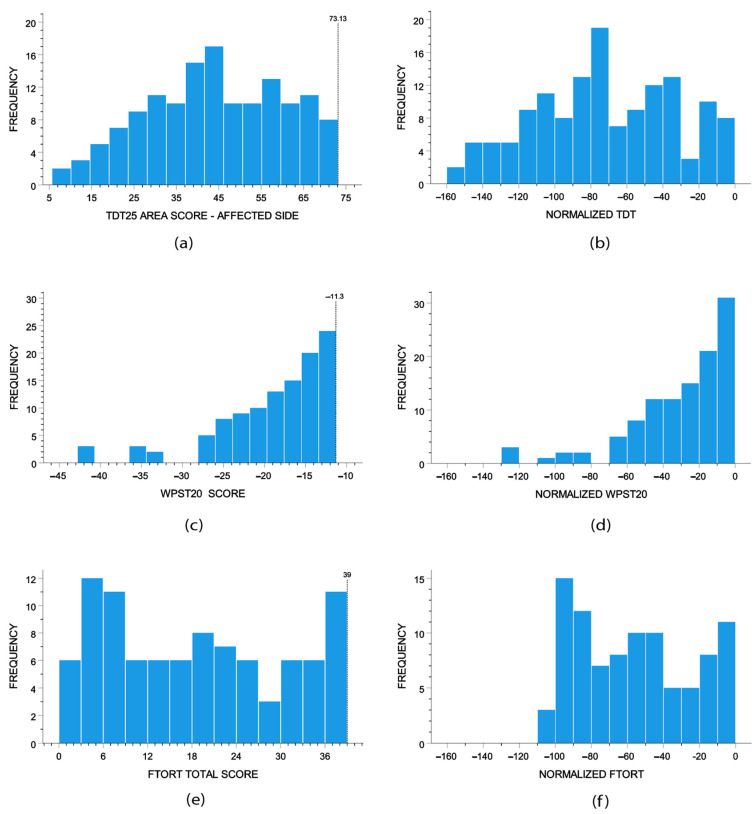
Histograms of untransformed raw scores (left panels; (**a**,**c**,**e**)) and normalized impaired scores (right panels; (**b**,**d**,**f**)) on the TDT25, WPST20 and fTORT collected at baseline from stroke survivors. The criterion of abnormality for the TDT25 is 73.13 PMA (**a**); for the WPST, 11.3 average error (**c**); and for the fTORT, 39 total score (**e**). Normalized impairment scores (**b**,**d**,**f**) start at zero.

**Figure 7 brainsci-13-00654-f007:**
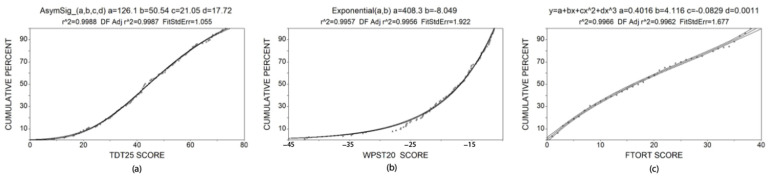
Cumulative distributions, fitted CDFs, and fit statistics for (**a**) TDT25 scores; (**b**) WPST20 scores; (**c**) fTORT scores. The fitted functions illustrate the difference in the rate of progression of percentile scores from 0% to 100%, with the TDT25 that conformed well to an asymmetric sigmoid function, in contrast to WPST20 percentiles progressing in line with an exponential function, while a cubic polynomial described well the fTORT progression. These CDFs reinforce the impression of dissimilar distribution shapes.

**Table 1 brainsci-13-00654-t001:** Demographic and clinical characteristics of pooled stroke sample for TDT, WPST, and fTORT across pooled samples.

Demographic and Clinical Information	Pooled Sample TDT (*n* = 174)	Pooled Sample WPST (*n* = 112)	Pooled Sample fTORT (*n* = 98)
*Study samples n =*			
^a^ Discriminative Validity	27	27	0
^b^ SENSe	44	37	48
^c^ IN_Touch	19	34	11
^d^ CoNNECT	45	14	39
^e^ NIH Toolbox	7	-	-
^f^ SENSe CONNECT	32	-	-
*Age* (*years*), *M* (*SD*)	56.7 (14)	55.4 (14.2)	58.1 (13.4)
*Gender, n%*			
Male	124 (71.3)	80 (71.4)	71 (72.4)
Female	50 (28.7)	32 (28.6)	27 (27.6)
*Time post-stroke* (*weeks*)			
Median (IQR)	42 (12.1–86.9)	25 (7.25–68.75)	44 (20–82)
*Lesion Level*, *n* (*%*)			
Cortical	74 (42.5)	50 (44.6)	45 (45.9)
Subcortical	53 (30.5)	28 (25)	30 (30.6)
Both	16 (9.2)	19 (17)	16 (16.3)
Unknown	31 (17.8)	15 (13.4)	7 (7.1)
*Stroke type*, *n* (*%*)			
Ischemic	97 (55.8)	74 (66.1)	73 (74.5)
Haemorrhage	50 (28.7)	23 (20.5)	25 (25.5)
Unknown	27 (15.5)	15 (13.4)	0
*Hemisphere affected*, *n* (*%*)			
Right	85 (48.9)	45 (40.2)	41 (41.8)
Left	86 (49.4)	66 (58.9)	55 (56.1)
Both	3 (1.7)	1 (0.9)	2 (2)
*Affected side*, *n* (*%*)			
Dominant	85 (48.9)	60 (53.6)	54 (55.1)
Non-dominant	88 (50.6)	48 (42.9)	44 (44.9)
Unknown	1 (0.6)	4 (4)	0

M, mean; SD; standard deviation; IQR interquartile range. ^a^ Discriminative Validity Study; ^b^ SENSe, Study of Effectiveness of Neurorehabilitation on Sensation; ^c^ IN_Touch, Imaging Neuroplasticity of Touch; ^d^ CoNNECT, Connecting Networks for Everyday Contact Through Touch; ^e^ NIH Toolbox, additional testing linked with National Institute of Health Toolbox Trial; ^f^ SENSe CONNECT Study.

**Table 2 brainsci-13-00654-t002:** Principal statistics for simulations of area scores of the TDT25 obtained from a biased insensate responder operating in seven states of bias.

Bias State:	None	Central (B = 50%)	Central (B = 66.67%)	Lateral (C = 50%)	Lateral (C = 66.67%)	Lateral (A = 50%)	Lateral (A = 66.67%)
Mean	33.31	36.25	39.04	31.41	29.47	32.36	31.45
Median	32.76	35.86	39.31	31.03	28.97	32.07	31.03
Standard Deviation	10.54	10.47	9.38	10.06	8.97	10.10	9.08
Coeff of variation	0.32	0.29	0.24	0.32	0.30	0.31	0.29
Skewness	0.17	0.12	0.02	0.20	0.25	0.12	0.20
Kurtosis	−0.09	−0.12	−0.12	−0.08	−0.01	−0.16	0.00
Percentiles							
5	16.55	19.31	23.45	15.17	15.17	15.86	16.90
25	25.86	28.97	32.76	24.48	23.10	25.17	25.17
50	32.76	35.86	39.31	31.03	28.97	32.07	31.03
75	40.34	43.45	45.52	38.28	35.17	39.31	37.24
95	51.38	54.14	54.83	48.60	44.83	49.31	46.90

## Data Availability

The raw clinical data are not publicly available due to planned further analyses. Data simulations may be requested from the corresponding author.

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
