# Peer review of "Calibration of Impairment Severity to Enable Comparison across Somatosensory Domains"

_brainsci, 2023, doi:10.3390/brainsci13040654_

Round 1
Reviewer 1 Report
Recent studies have spent more attention to post-stroke somatosensory impairments, which are often overshadowed by their motor counterparts. But the somatosensory system itself comprises multiple sensory domains (touch, proprio-, thermal, etc), and direct comparison of the impairment level among those poses practical challenges. Common practice in cross-modality comparison can include standardization of scores, but this method also depends on the underlying distribution of the data.
In the submitted manuscript, the authors addressed this problem by proposing a method for finding comparable locations in the psychophysical score ranges, covering from just noticeably impaired healthy adults to maximally impaired patients. For this manuscript, the authors picked 3 somatosensory domains: texture discrimination, wrist proprioception, and haptic object recognition. For each domain, comparable ranges of impairment (from hardly impaired to extremely impaired conditions) were determined using these estimations and previously identified values for the domain impairment threshold. The authors found that the shapes of distribution of scores in the 3 domains investigated were found to be non-homogenous.
Overall, this is a meaningful study to report. Shapes of the distribution of scores reported are informative. I had a difficult time finding fault in the author’s work. The authors were meticulous about presenting the findings in a logical order that was easy to follow. They provided clear objectives and created a well-written manuscript. As a researcher interested in sensory impairment post-stroke, I look forward to citing this text myself.
Some feedback/suggestions for consideration:
1) Can the authors briefly discuss how they chose the 3 modalities? Why is it focusing on distal limb (end effector)?
2) What do the authors think about using the more standardized Em-NSA and RASP ax. for the cross-domains comparison? Perhaps can include this in the Discussion.
3) Section 2.4.1 to 2.4.3: Is ‘Simulation’ more appropriate than ‘Evaluation’?
4) Cosmetics:
Figure 4 is blurry, possible to change to plots with more legible text?
Figure 6 (a) and (b) textboxes cover the axis label.
Author Response
Please see the attachment that outlines response to reviewers, tracked changes in revised manuscript and pdfs of updated figures.

Reviewer 2 Report
This is good manuscipt for neurologist and therapist.
- Line 34, 41, References are needed to support this sentense.
- In order to improve your manuscript, please statistically analyze the onset time and mention it.
- What are the limitation of your study.
Author Response

(The authors gave the same response as above.)
